# Functional Bayesian Neural Networks for Model Uncertainty Quantification

## Abstract

In this paper, we extend the Bayesian neural network to functional Bayesian neural network with functional Monte Carlo methods that use the samples of functionals instead of samples of networks' parameters for inference to overcome the curse of dimensionality for uncertainty quantification. Based on the previous work on Riemannian Langevin dynamics, we propose the stochastic gradient functional Riemannian dynamics for training functional Bayesian neural network. We show the effectiveness and efficiency of our proposed approach with various experiments.

## 1    Introduction

Though deep neural networks have achieved tremendous success in various domains recently, there still exists some fundamental issues largely unclear and unsolved. One of them is that the training of neural networks usually ignores the uncertainty over the parameters since typically maximum a posteriori (MAP) is adopted for learning the parameters and only point estimates could be obtained. This problem can cause neural networks to make overconfident predictions, particularly in the region poorly covered by training data or far away from the underlying data manifold. Besides, the ignorance of model uncertainty also make neural networks vulnerable to adversarially perturbed data, so-called "adversarial examples" (Szegedy et al., 2013; Gal & Smith, 2018).

The Bayesian perspective provides a principled way of accounting for the uncertainty of model's prediction by integrating over the posterior distribution over parameters. In the scenario of neural networks, Bayesian neural networks (MacKay, 1992; Hinton & Van Camp, 1993; Neal, 1995) was proposed. However, for modern neural networks, the integral over posterior distribution is highly intractable due to the large network size and complex multi-modality.

There are mainly two groups of approaches for handling the intractability of the complex high-dimensional distribution. The first one is to employ various Markov Chain Monte Carlo (MCMC) (Neal (1993); Max & Whye (2011); Chen et al. (2014); Shang et al. (2015); Ye et al. (2017)) to obtain the samples from the distribution; and then the Monte Carlo approximation can be used. However, the high dimensionality and a large number of modes in Bayesian neural networks prohibit the scaling of current MCMC methods, i.e. the mixing is extremely time-consuming. The other method is variational inference, typically approximating the original complex posterior distribution with independent Gaussian distributions over each weight in networks (Graves (2011); Hernández-Lobato & Adams (2015); Blundell et al. (2015); Gal & Ghahramani (2016)). The limitation of this strategy is that the introduced approximation might underestimate the posterior when optimizing the variational lower bound.

In this paper, we extend the Bayesian neural network to functional Bayesian neural network for MCMC methods. Instead of sampling a single parameter each step, we propose to sample a function which can represents the posterior distribution each step. Based on the previous work for functional Hamiltonian method, we extend a previous work on Riemannian Langevin dynamics and propose its functional extension with mean-field Gaussian approximation.

## 2    Functional Bayesian Neural Network

In this section, we will first explain the preliminary of Bayesian neural network and then extend it into the functional space.

Neural networks are most commonly trained in a maximum a posteriori (MAP) setting, where only point estimates of the network parameters can be obtained. This ignores any uncertainty about the parameters that often result in overconfident predictions, especially in regimes that are weakly covered by training data or far away from the data manifold. Bayesian neural networks Neal (1995) incorporates neural networks into the Bayesian framework for accounting for the parameter uncertainty through modeling the posterior distribution.

Firstly, one need to provide a prior distribution over the weights, $p_0(\boldsymbol{\theta}) = \mathcal{N}(\mathbf{0}, \sigma_0^2 \boldsymbol{I})$, where $\sigma_0^2$ is the variance magnitude. Assuming the likelihood function has the form,

$$p(\mathbf{y}_i|\mathbf{x}_i, \boldsymbol{\theta}) = \mathcal{N}\left(\mathbf{y}_i|u(\mathbf{x}_i; \boldsymbol{\theta}), \sigma^2 \mathbf{I}\right), \tag{1}$$

where $u(\mathbf{x}; \boldsymbol{\theta})$ represents the output of the neural network. Then the posterior of the weights $\boldsymbol{\theta}$ is $p(\boldsymbol{\theta}|\mathcal{D}) \propto \prod_{i=1}^{N} p(\boldsymbol{y}_i|\boldsymbol{x}_i, \boldsymbol{\theta})p_0(\boldsymbol{\theta})$.

The uncertainty of the model, typically formulated as the expectation of a specific statistics $g(\boldsymbol{\theta}; \mathbf{x}, \mathbf{y})$. could be computed based on the posterior distribution of the weights,

$$\mu_g = \mathbb{E}[g(\boldsymbol{\theta}; \mathbf{x}, \mathbf{y})] = \int g(\boldsymbol{\theta}; \mathbf{x}, \mathbf{y})p(\boldsymbol{\theta}|\mathcal{D})\mathrm{d}\boldsymbol{\theta}. \tag{2}$$

Due to the analytic intractability of the integral, the expectation of $g(\boldsymbol{\theta}; \boldsymbol{x}, \boldsymbol{y})$ is estimated using Monte Carlo integration

$$\mu_g \approx \frac{1}{T} \sum_{i=1}^{T} g(\boldsymbol{\theta}_i; \boldsymbol{x}, \boldsymbol{y}), \tag{3}$$

where $\{\boldsymbol{\theta}_i\}_{i=1}^{M}$ is drawn from the posterior $p(\boldsymbol{\theta}|\mathcal{D})$.

To derive the functional Bayesian neural networks, we first define a functional $f : \boldsymbol{\theta} \to [0, 1]$ which maps a parameter setting $\boldsymbol{\theta}$ to a probability, or more formally, we define a separable Hilbert space $\mathcal{H}$ with inner product and norm operation and the functional $f : \boldsymbol{\theta} \to [0, 1]$ is in $\mathcal{H}$. Note that $f$ is a probability distribution of weights $\theta$. Similar to Bayesian neural networks, suppose we get posterior distribution of functionals given the data: $p(f|\mathcal{D}) \propto \prod_{i=1}^{N} p(\boldsymbol{y}_i|\boldsymbol{x}_i, \boldsymbol{f})p_0(f)$, where $p_0(f)$ is the prior distribution of $f$, the uncertainty of the model can be formulated as the expectation of a specific statistics $g(\boldsymbol{\theta}; \mathbf{x}, \mathbf{y})$:

$$\mu_g = \mathbb{E}[g(f; \mathbf{x}, \mathbf{y})] = \int g(f; \mathbf{x}, \mathbf{y})p(f|\mathcal{D})\mathrm{d}f, \tag{4}$$

where $g(f; \mathbf{x}, \mathbf{y}) = \int g(\boldsymbol{\theta}; \mathbf{x}, \mathbf{y})f(\boldsymbol{\theta})d\boldsymbol{\theta}$ is the expectation of the $g(\boldsymbol{\theta}; \mathbf{x}, \mathbf{y})$ given the distribution functional $f$. Due to the analytic intractability of the integral, similarly, the expectation of $g(\boldsymbol{\theta}; \boldsymbol{x}, \boldsymbol{y})$ can be estimated using a two-loop Monte Carlo integration,

$$\mu_g \approx \frac{1}{T} \sum_{i=1}^{T} \sum_{j=1}^{M} g(\boldsymbol{\theta}_{ij}; \boldsymbol{x}, \boldsymbol{y}), \text{Sample } \boldsymbol{\theta}_{ij} \text{ from } f_i, \tag{5}$$

where we sample $M$ parameters from each posterior $f_i$ and $T$ posteriors is used for computing the expectation. Through this formulation, we can perform inference with functional Bayesian neural networks. Note that with the same number of iterations $T$, the functional Bayesian neural network can have $M$ times more samples than Bayesian neural networks. We provide a functional sampling example in section 5.1 for illustration.

In the next parts, we will propose the stochastic gradient functional Riemannian Langevin dynamics (SGFuncRLD) for training functional Bayesian neural network.

## 3 FUNCTIONAL RIEMANNIAN LANGEVIN DYNAMICS

Beskos et al. (2011) first introduced the concept of functional Monte Carlo method on Hilbert space with (second-order) Langevin dynamics. Beskos *et al.* proved that if the target distribution $U(\boldsymbol{\theta})$ can be represented by the sum of weighted sequence of functionals $f$ with the coefficients of functionals

decreasing faster than a polynomial rate, along with other necessary assumptions, sampling the functionals can be equivalent to sampling the posterior $U(\boldsymbol{\theta})$ directly. However, the HMC method cannot adapt to the local geometry and need to tune parameters carefully and in real practice, tuning hyper-parameter can be costly. We want the training algorithm to be scalable and robust to different hyper-parameter settings, we thus extend the Riemannian Langevin dynamics to functional space to make the functional MCMC method more robust.

Similar to Beskos et al. (2011), we represent the posterior $U(\boldsymbol{\theta})$ with its finite approximation $U_D(\boldsymbol{\theta}) = \sum_i^D \boldsymbol{\lambda}_i u_i$. We denote the parameter of $U_D(\boldsymbol{\theta})$ to be $\boldsymbol{\lambda} \in \mathcal{R}^D$ thus given $\boldsymbol{\lambda} \in \mathcal{R}^D$, we have a mapping from $\boldsymbol{\lambda}$ to $U_D(\boldsymbol{\theta})$. Note that this mapping is continuous and bounded. Then by sampling $\boldsymbol{\lambda}$, we sample a functional $f$ equivalently. The Riemannian Langevin dynamics on the functional space can thus be written as:

$$\mathrm{d}\boldsymbol{\lambda} = \mathbf{b}(\boldsymbol{\lambda})\mathrm{d}t + \sqrt{2G(\boldsymbol{\lambda})}\mathrm{d}\mathbf{W} \tag{6}$$

where $\mathrm{d}\mathbf{W}$ represents the standard Brownian motion, and the drift force $\mathbf{b}(\boldsymbol{\lambda}, \alpha)$ has the following form,

$$\mathbf{b}(\boldsymbol{\lambda}) = -G(\boldsymbol{\lambda})\frac{\partial U(\boldsymbol{\lambda})}{\partial \boldsymbol{\lambda}} - \xi(\boldsymbol{\lambda}) \tag{7}$$

$$\xi_i(\boldsymbol{\lambda}) = -\sum_{j=1}^d \frac{\partial}{\partial \boldsymbol{\lambda}_j} G_{ij}(\boldsymbol{\lambda}), \tag{8}$$

where $G_{ij}$ is the Riemannian information metric. Similar to its counterpart defined in real number space, we have the following theorem,

**Theorem 1.** $p(\boldsymbol{\lambda}) \propto \exp\left(-U(\boldsymbol{\lambda})\right)$ *is a stationary distribution of the dynamics of Eq. (6), if $G(\boldsymbol{\lambda})$ is a positive semidefinite matrix.*

*Proof.* The Fokker-Planck equation of the dynamics in Eq. (6),

$$\partial_t \rho(\boldsymbol{\lambda}, t) = -\sum_i \frac{\partial}{\partial \boldsymbol{\lambda}_i} \left[ \sum_j b_j(\boldsymbol{\lambda}, \alpha)\rho(\boldsymbol{\lambda}, t) \right] + \sum_{i,j} \frac{\partial^2}{\partial \boldsymbol{\lambda}_i \boldsymbol{\lambda}_j} 2G_{ij}\rho(\boldsymbol{\lambda}, t) \tag{9}$$

We insert $\mathbf{b}(\boldsymbol{\lambda}) = -\mathbf{G}(\boldsymbol{\lambda})\frac{\partial U(\boldsymbol{\lambda})}{\partial \boldsymbol{\lambda}} - \xi(\boldsymbol{\lambda})$ into the above equation, the Fockker-Planck equation can be simplified as,

$$\partial_t \rho(\boldsymbol{\lambda}, t) = \sum_i \frac{\partial}{\partial \boldsymbol{\lambda}_i} \left[ \sum_j \{G_{ij}\frac{\partial U(\boldsymbol{\lambda})}{\partial \boldsymbol{\lambda}_j} - \frac{\partial G_{ij}}{\partial \boldsymbol{\lambda}_j}\}\rho(\boldsymbol{\lambda}, t) \right] + \sum_{i,j} \frac{\partial^2}{\partial \boldsymbol{\lambda}_i \boldsymbol{\lambda}_j} 2G_{ij}\rho(\boldsymbol{\lambda}, t) \tag{10}$$

When $\rho(\boldsymbol{\lambda}, t) = p(\boldsymbol{\lambda})$, the stationary distribution will be reached ($\partial_t \rho(\boldsymbol{\lambda}, t) = 0$). Then we observe that right hand side of the Fokker Planck equation vanishes. Thus, when $p(\boldsymbol{\lambda}, t) \propto \exp(-U(\boldsymbol{\lambda}))$, $\partial_t \rho(\boldsymbol{\lambda}, t) = 0$. $\square$

To simulate Functional Riemannian Langevin dynamics for neural networks, we use the following approximate computation for stochastic functional Riemannian Langevin dynamics (SGFuncRLD) for neural networks.

## 4 APPROXIMATE COMPUTATION

### 4.1 FINITE FUNCTIONAL APPROXIMATION

We use the mean-field Gaussian approximation to approximate $U(\boldsymbol{\theta})$. Mean-field Gaussian approximation has been widely used for neural networks and shows good performance for Bayesian inference for neural networks, such as in variational inference (Blei et al. (2017)). We use this approximation and thus $U(\boldsymbol{\theta})$ is represented in a simple one-line form,

$$p(\boldsymbol{\theta}|D) \approx p(f) = \prod_{i=1}^N p(\boldsymbol{\theta}_i|D) \tag{11}$$

where $p(\boldsymbol{\theta}_i|D) = \mathcal{N}(\mu_i, \sigma_i|D)$ and $\mathcal{N}(\mu_i, \sigma_i)$ is a normal distribution with mean $\mu_i$ and standard deviation $\sigma_i$. Note that unlike variational inference, we do not have the problem of zero forcing or zero avoiding (Murphy (2013)).

## 4.2 Stochastic computation approximation

We use the Fisher information metric as the Riemman information metric (RIM) as in the previous work (Li et al., 2016):

$$G(\boldsymbol{\lambda}) = \mathbb{E}\left[\frac{\partial U(\boldsymbol{\lambda})}{\partial \boldsymbol{\lambda}}\frac{\partial U(\boldsymbol{\lambda})}{\partial \boldsymbol{\lambda}}^T\right] \tag{12}$$

We only use the diagonal part of the RIM and set the remaining part to be zero for computationally purpose,

$$G_{ii}(\boldsymbol{\lambda}) = \mathbb{E}\left[\left(\frac{\partial U(\boldsymbol{\lambda})}{\partial \boldsymbol{\lambda}_i}\right)^2\right] \tag{13}$$

We approximate the gradients on the dataset with the smoothed gradients on mini-batches for scalable computation. Besides, as we only need to approximate form of RIM, we do not consider $\xi(\boldsymbol{\theta})$ for scalable computation on big data.

We use the gradient computed on the $k$-th mini-batch $\{\mathbf{x}_{k_1}, \ldots, \mathbf{x}_{k_m}\}$ to approximate the gradient,

$$\tilde{U}(\boldsymbol{\lambda}) = -\frac{N}{m}\sum_{j=1}^{m}\log p(\mathbf{x}_{k_j}|\boldsymbol{\lambda}) - \log p_0(\boldsymbol{\lambda}). \tag{14}$$

However, previous work in optimization literature shows that the smoothed and bias-corrected form of gradient shows more stable performance than this unbiased stochastic approximation when scaled to large datasets (Kingma & Ba (2014)), we then approximate the gradient with smoothed and bias-corrected version,

$$\mathbf{m}_t = \beta_1\mathbf{m}_{t-1} + (1 - \beta_1)\nabla_{\boldsymbol{\lambda}}\tilde{U}(\boldsymbol{\lambda}_t) \tag{15}$$
$$\mathbf{m}_t = \mathbf{m}_t/(1 - \beta_1^t) \tag{16}$$

where $\mathbf{m}_t$ is the smoothed gradient at step $t$, $\beta_1$ is the smoothing coefficient. We also apply the same technique for computing the RIM.

$$\mathbf{G}_t(\boldsymbol{\lambda}) = \beta_2\mathbf{G}_{t-1}(\boldsymbol{\lambda}) + (1 - \beta_2)(\nabla_{\boldsymbol{\lambda}}\tilde{U}(\boldsymbol{\lambda}_t))^2 \tag{17}$$
$$\mathbf{G}_t(\boldsymbol{\lambda}) = \mathbf{G}_t(\boldsymbol{\lambda})/(1 - \beta_2^t) \tag{18}$$

where $\mathbf{G}_t$ is the smoothed gradient at step $t$, $\beta_2$ is the smoothing coefficient.

## 4.3 Stochastic gradient functional Riemannian dynamics

With these approximations, we have the algorithm in Algorithm 1. After we derive samples of the coefficients $\boldsymbol{\lambda}_i$ for the mean-field Gaussian distribution, we can use Eq. 5 for inference.

## 5 Experiments

In this section, we will first illustrate how functional MCMC method works with functional Hamiltonian dynamics and Hamiltonian dynamics on a double-well potential function. Then we will compare our proposed method with other methods including stochastic gradient Langevin dynamics (Max & Whye, 2011) (SGLD), stochastic gradient Hamiltonian Monte Carlo (Max & Whye, 2011)(SGHMC), preconditioned stochastic gradient Langevin dynamics (Li et al., 2016) (pSGLD) in various examples with neural networks. In the following experiments for neural networks, we set the number of (functional) samples $T$ to be 5, and for functional Bayesian neural network, we set the number of samples from each functional $M$ to be 5. For SGFuncRLD, we set $\beta_1$ to be 0.9 and $\beta_2$ to be 0.999. We implement the experiments with Tensorflow and Tensorflow-probability.

---

Algorithm 1: SGFuncRLD for training Bayesian neural networks

---

1: **Input:** $\beta_1, \beta_2, \eta$.
2: Initialize $\boldsymbol{\lambda}_0, \mathbf{m}_0, \mathbf{G}_0$.
3: **for** $t = 1, 2, \ldots$ **do**
4:      Randomly sample a minibatch of the dataset to obtain $\tilde{U}(f_{\boldsymbol{\lambda}_t})$:
5:      Sample $\boldsymbol{\epsilon}_t \sim \mathcal{N}(\mathbf{0}, \mathbf{I})$ from the standard Gaussian distribution ;
6:      Compute the smoothed gradient $\mathbf{m}_t$;

$$\mathbf{m}_t = \beta_1 \mathbf{m}_{t-1} + (1 - \beta_1)\nabla_{\boldsymbol{\lambda}}\tilde{U}(\boldsymbol{\lambda}_t), \qquad \mathbf{m}_t = \mathbf{m}_t/(1 - \beta_1^t)$$

7:      Compute the smoothed RIM $\mathbf{G}_t(\boldsymbol{\theta})$;

$$\mathbf{G}_t(\boldsymbol{\lambda}) = \beta_2 \mathbf{G}_{t-1}(\boldsymbol{\lambda}) + (1 - \beta_2)(\nabla_{\boldsymbol{\lambda}}\tilde{U}(\boldsymbol{\lambda}_t))^2, \qquad \mathbf{G}_t(\boldsymbol{\lambda}) = \mathbf{G}_t(\boldsymbol{\lambda})/(1 - \beta_2^t)$$

8:      Update the parameter;

$$\boldsymbol{\lambda}_t = \boldsymbol{\lambda}_{t-1} - \mathbf{G}_t(\boldsymbol{\lambda})^{-\frac{1}{2}}\eta\mathbf{m}_t + \sqrt{2\mathbf{G}_t(\boldsymbol{\lambda})^{-1}\eta}\boldsymbol{\epsilon}_t$$

9: **end for**

---

## 5.1 ILLUSTRATION OF FUNCTIONAL MCMC

To illustrate how functional MCMC methods work, we show an example of sampling double-well potential function. Double-well potential function is widely used for evaluating MCMC methods, the target distribution function used is: $U(\theta) = -2\theta^2 + 0.2\theta^4$, similar to the one used in Chen et al. (2014). Note that the sampling iteration is set to be 10 instead of $80000 \times 50$ in the original example (Chen et al. (2014)) to test the methods' ability to efficiently sample the parameter space in a limited amount of time. For Functional MCMC, we set the initialization functional to be a standard normal distribution. We use the Hamiltonian dynamics for the Functional MCMC method-FuncHMC, which is the functional extension of the widely-used Hamiltonian Monte Carlo (HMC). For fair comparison, we use the same parameter for FuncHMC and HMC (learning rate is 0.2, momentum is 0.5). We show the sampled functional from FuncHMC at each step and HMC's samples in Figure 1 (Left). From Figure 1 (Left), we can observe that different from HMC that can only generate one single sample of parameter at each time, FuncHMC generates one sample of distribution function at each time. Even with very few iterations, FuncHMC is able to sample the multi-modal distribution while HMC's samples are concentrated near $\theta = 1$. This property can help overcome the curse of dimensionality when sampling for high dimension distributions especially for Bayesian neural networks.

To compare quality of samples generated by FuncHMC or HMC, we use the KLIEP [1] to estimate the ratio distribution of benchmark samples to FuncHMC samples or HMC samples. We use HMC with a learning rate of 0.02 and a momentum of 0.5 and run it for $10^6$ iterations and sample every 100 iterations which gives a total of $1e^4$ samples as the benchmark samples.

From Figure 1, we can observe that ratio distribution of FuncHMC varies close to the line of $y = 1$. Though HMC achieves good approximation close to $\theta = 1$, HMC fails to correlate well with the benchmark samples within a broader range. For FuncHMC, we generate 100 samples from each posterior functional sampled and this gives us 1000 samples in total from 10 iterations. The plot of the ratio distribution is shown in Figure 1 (Right). From Figure 1 (Right), we can observe that ratio distribution of FuncHMC varies closely to the black horizontal line $y = 1$. Besides, HMC can achieve good approximation close to $\theta = 1$, but it fails to correlate well with the benchmark samples across the distribution range in a limited amount of steps.

Next, we will compare our proposed method SGFuncRLD with other popular SG-MCMC methods for Bayesian neural networks on uncertainty estimation.

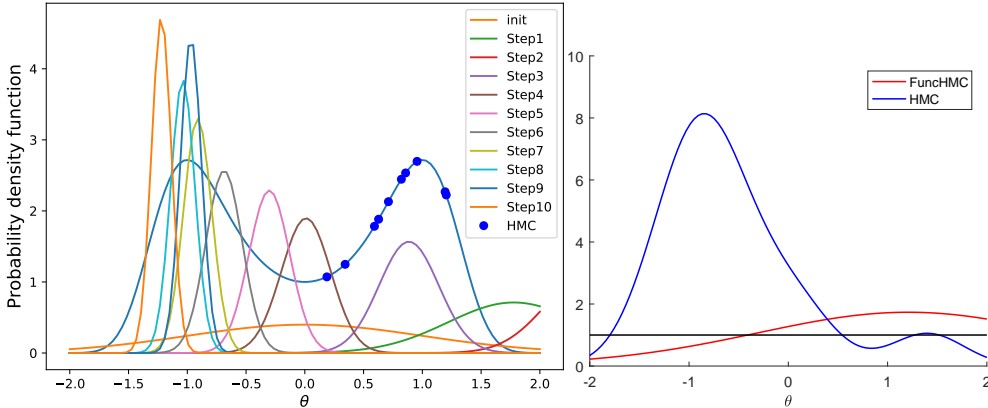

Figure 1: **Left:** Sampled functionals by FuncHMC and samples generated by HMC. **Right:** Ratio distribution of benchmark to FuncHMC or HMC (The closer to the black line $y = 1$ the better).

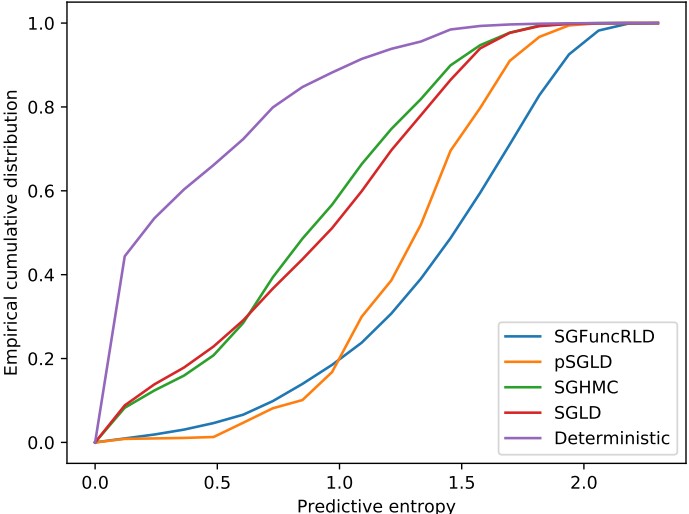

Figure 2: Empirical cumulative distribution function of predictive entropy (The lower the better).

## 5.2 UNCERTAINTY ESTIMATION ON OUT-OF-DISTRIBUTION DATA

Accurate uncertainty estimation on unseen data is key to many safety-critical applications, such as autonomous driving and disease diagnosis. To measure the networks' ability on uncertainty estimation for out-of-distribution data, we use a similar experiment setting as in Ritter et al. (2018). We train Bayesian neural networks and the functional Bayesian neural network on the MNIST dataset and use the trained models to predict on the unseen NoMNIST dataset[2] that consists of images of letters from "A" to "J" with various fonts. We compare the entropy of predictions on the NoMNIST dataset. The minimum value of the predictive entropy is 0 and the maximum value of the predictive entropy is $\log(10) \approx 2.30$. We use the neural network architecture of conv (6 filters with a kernel size of 5)-pool-conv (16 filters with a kernel size of 5)-pool-conv (120 filters with a kernel siz of 5)-fully connected (84)-fully connected (10), which is the same as in the Tensorflow probability example [3]. We train the network for 5000 iterations and use a batch size of 128, which is the same as in the official example. In the training phase, we set the learning rate of SGFuncRLD, pSGLD and Adam(Deterministic method) to be 0.001, SGLD and SGHMC to be 0.1, and the momentum for

---

[1]Sugiyama et al. (2008) http://www.ms.k.u-tokyo.ac.jp/software.html

[2]https://www.kaggle.com/lubaroli/notmnist

[3]https://github.com/tensorflow/probability/blob/r0.3/tensorflow_probability/examples/bayesian_neural_network.py

SGHMC to be 0.1. In the test phase, we draw five samples (functionals) for MCMC methods such as SGFuncRLD, pSGLD, SGHMC and SGLD; For SGFuncRLD, we further draw five samples for each functional; For deterministic training method, one sample is used. We average the predicted probability from samples as the final predicted probability. The empirical cumulative distribution function of predictive entropy on NoMNIST dataset is shown in Figure 2. From Figure 2, we can observe that all MCMC methods can achieve better uncertainty estimation performance than the deterministic method which indicates that Bayesian neural network is necessary for safety-critical applications requiring accurate uncertainty estimation. SGFuncRLD achieves the best performance by predicting the unseen examples with high entropy.

## 6 UNCERTAINTY ESTIMATION ON ADVERSARIAL ATTACK DATA

Though neural networks have achieved great success in many applications, they are found to be vulnerable to maliciously manipulated perturbation over the original data (Szegedy et al., 2013). To test the neural networks' uncertainty estimates to adversarial attack, we generate the adversarial examples from one instance of each Bayesian neural network trained by different MCMC methods and compute the predictive entropy of the adversarial attack examples from other instances to quantify the uncertainty of the neural network's predictions. We use the same architecture and configuration as the previous experiment but test with MNIST data instead. We use the fast gradient sign method (FGSM) to generate the adversarial data with different attack strength-epsilon. We plot the predictive entropy and accuracy on the attacked data in Figure 3. From Figure 3, we can observe that with the increase of attack strength epsilon, the predictive entropy of SGFuncRLD increases quickly while the accuracy decreases smoothly. When the FGSM attack epsilon is below 0.05, the accuracy remains close to 98% . When the FGSM attack epsilon is above 0.05, the predictive entropy increases quickly while the accuracy decreases smoothly. Note that for other methods, when the attack epsilon is larger than 0.2, the predictive entropy does not increase, which might be dangerous in real practice as the strongly-perturbed data still have high confidence for Bayesian neural network. This indicates that SGFuncRLD trained Functional Bayesian neural network can achieve the best performance on uncertainty estimation on adversarial attack data. We speculate that this is because sampling in the high dimensional space is hard for common MCMC method and the proposed SGFuncRLD can explore the parameter space much more efficiently to capture the multi-modality.

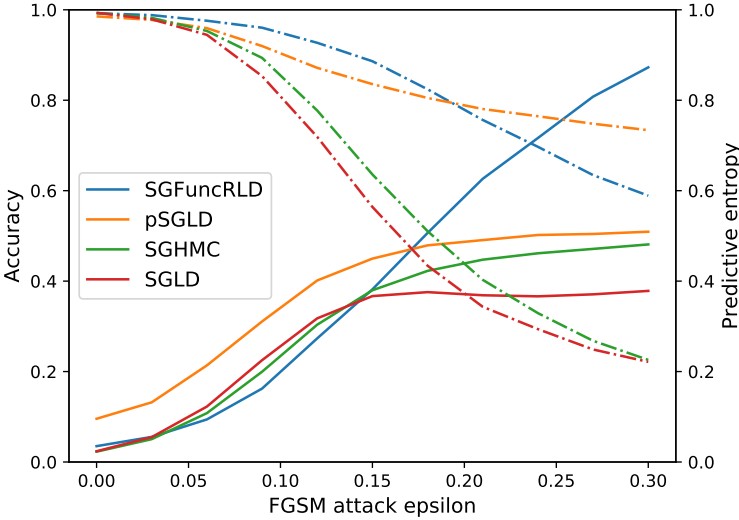

Figure 3: Uncertainty estimation on adversarial data.

Next, we will test our proposed method's generalization abilities on a practical application.

### 6.1 TRAFFIC SIGN RECOGNITION

To test methods' generalization abilities, we use the German traffic sign recognition benchmarks (GTSRB) dataset. The GTSRB dataset consists of 39209 training images and 12630 test images. We

further split the training images into two part-34799 images for training, 4410 images for validation. Then We use a neural network with similar architecture as LeNet Lecun et al. (1998). We use the gray version of the traffic sign images and do the image local normalization and normalize the images for pre-processing the data. The code is adapted from the repository [4]. The network architecture is conv (6 filters with a kernel size of 5)-pool-conv (16 filters with a kernel size of 5)-pool-fully connected (128)-fully connected (43). We do a grid search to determine the best parameter for each method. The best parameter setting for each method is: SGFuncRLD(learning rate is 0.001), pSGLD(learning rate is 0.0001), SGHMC(learning rate is 0.1, momentum is 0.6), SGLD(learning rate is 0.1). The training and validation curves are shown in Figure 4. From Figure 4, we can observe that the SGLD converges faster among all methods with SGFuncRLD only slower than SGLD. However, different from all other methods, the training and validation curves are more smooth for SGFuncRLD. All methods can achieve similar results after mixing. We speculate that this is because the landscape of loss function in functional space is smoother than the loss function in parameter space because of the mean-field Gaussian approximation. We test the trained models on the test dataset. As we observe that there is some small numerical difference for each run, we run the experiment five times and get the mean and the standard deviation of test accuracy as shown in Table 1. From Table 1, we can observe that the SGFuncRLD can achieve the best generalization performance on the large test dataset.

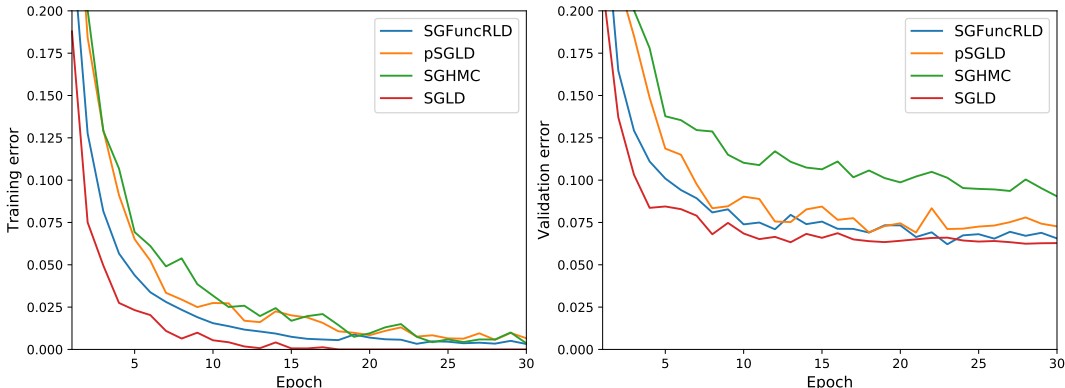

Figure 4: **Left:** Training curve. **Right:** Validation curve.

| Method | Test accuracy |
|---|---|
| **SGFuncRLD** | **91.67% ± 0.28%** |
| pSGLD | 90.17% ± 0.39% |
| SGHMC | 88.35% ± 0.20% |
| SGLD | 91.10% ± 0.11% |

Table 1: Test accuracy of CNN on traffic sign recognition.

## 7 CONCLUSION

In this work, we extend the Bayesian neural networks to functional Bayesian neural networks for model uncertainty quantification. We propose a novel stochastic functional Riemannian dynamics for training functional Bayesian neural networks. We show the superiority of functional Bayesian neural networks over various examples.

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
