# OpenReview forum: "Functional Bayesian Neural Networks for Model Uncertainty Quantification"
_ICLR.cc/2019/Conference_

### Official Review · AnonReviewer2 · 2018-11-02
**A functional version of Riemannian Langevin dynamics is used in order to perform inference with Bayesian neural networks. It is not quite convincing due to a lack of effort in explaining the approach.**

**Rating:** 5
**Confidence:** 2

**Review:**

The idea of extending  Riemannian Langevin dynamics to functional spaces is elegant, however it is extremely hard to follow the proposed method as details are kept to a minimum. The finite approximation of the posterior distribution is a function of the parameters theta, however it displays parameters lambda. The couple of sentences: "Then by sampling λ, we sample a functional f equivalently. The Riemannian Langevin dynamics on the functional space can thus be written as: (6)" come without a single explanation.

Minor comments
* Max and Whye is the casual version for reference Welling and Teh.
* proper nouns in References should be capitalized

---

### Official Review · AnonReviewer1 · 2018-11-03
**Unclear writing and contributions**

**Rating:** 4
**Confidence:** 4

**Review:**

This paper considers a new learning paradigm for Bayesian Neuron Networks (BNN): learning distribution in the functional space, instead of weight space. A new SG-MCMC variant is proposed in Algorithm 1, and applied to sampling in a
"functional space". The approach is demonstrated on various tasks.

Quality: Low, due to the low clarity detailed below.


Clarity: I do not fully follow the core algorithm:  The posterior is U_D(\theta) = \sum_{i=1}^D  \lambda_i * u_i, where  \lambda_i is represented as MCMC samples,  what is u_i then? I guess u_i is defined in (2), which is approximated in (3) if weight sample is used. However, how is u_i represented in the functional approach? I guess it is similar to the weight-based approach. If this is true, how could we distinguish between a functional approach and weight-based approach?

The proposed SGFuncRLD is essentially Adam plus Gaussian noise, but performed in a so-called "functional space"? It is therefore not surprise to me that SGFuncRLD performs better than pSGLD (RMSprop plus Gaussian noise), just as Adam performs better than RMSprop. If we only focus on the new SG-MCMC approach itself, the authors need to justify: (1) the smoothed gradient is an unbiased gradient estimator, how does it effect convergence? Does it guarantee to  true posterior? this should be done in theory. (2)  The SGFuncRLD  algorithm itself is the same with pSGLD except the smoothed gradient part.  This makes  the clear comparison even important. Does SGFuncRLD  perform better just because the proposed smoothed gradient, or because the sampling is done in the functional space?

My suggestions: Please disentangle the contributions clearly. There are two things: (1) smooth gradient, (2) sampling in a functional space. Which one really contributes the performance improvement?

To demonstrate (1),  the authors could at least conduct on a toy distribution, to demonstrate the difference with pSGLD, regardless it is to the functional space or the weight space.
To demonstrate (2), the authors  could apply the same SG-MCMC variant to the functional space and to the weight space, and see the difference.

Originality: To me, the idea of learning uncertainty of BNN in the functional space appeared in Prof.  Yee Whye Teh's NIPS 2017 presentation. The motivation in his presentation is very clear. However, how to implement this abstract idea in practice is unclear yet. This submission is the first attempt. However, I am concerned about the real contribution.

Significance: It is a very interesting research direction. The paper could have been significant if every part is clearly motivated and demonstrate. At this point, I am not fully convinced.

---

### Official Review · AnonReviewer3 · 2018-11-03
**An interesting idea plagued by flaws in presentation, inconsistent notation, and lack of critical experiments**

**Rating:** 3
**Confidence:** 3

**Review:**

The authors propose an approximate MCMC method for sampling a posterior distribution of weights in a Bayesian neural network.  They claim that existing MCMC methods are limited by poor scaling with dimensionality of the weights, and they propose a method inspired by HMC on finite-dimensional approximations of measures on an infinite-dimensional Hilbert space (Beskos et al, 2011).  In short, the idea is to use a low dimensional approximation to the parameters (i.e. weights) of the neural network, representing them instead as a weighted combination of basis functions in neural network parameter space.  Then the authors propose to use HMC on this lower dimensional representation.  While the idea is intriguing, there are a number of flaws in the presentation, notational inconsistencies, and missing experiments that prohibit acceptance in the current form.

The authors define a functional, f: \theta -> [0, 1], that maps neural network parameters \theta to the unit interval.  They claim that this function defines a probability distribution on \theta, but this not warranted.  First, \theta is a continuous random variable and its probability density need not be bounded above by one; second, the authors have made no constraints on f actually being normalized.

The second flaw is that the authors equate a posterior on f given the data with a posterior on the parameters \theta themselves.  Cf. Eq 4 and paragraph above.  There is a big difference between a posterior on parameters and a posterior on distributions over parameters.   Moreover, Eq. 5 doesn't make sense: there is only one posterior f; there are no samples of the posterior.

The third problem appears in the start of Section 3, where the authors now call the posterior U(theta) instead of f.  They make a finite approximation of posterior U(\theta) = \sum_i \lambda_i u_i, which is inconsistent with Beskos et al.  I believe the authors intend to use a low dimensional approximation to \theta rather than its posterior U(\theta).  For example, if \theta = \sum_i \lambda_i u_i for fixed basis functions u_i, then you can approximate a posterior on \theta with a posterior on \lambda.

The fourth, and most important problem, is that the basis functions u_i are never defined.  How are these chosen? Beskos et al use the eigenfunctions of the Gaussian base measure \pi_0, but no such measure exists here.  Moreover, this choice will have a substantial impact on the approximation quality.

There are more inconsistencies and notational problems throughout the paper.  Section 4.1 begins with a mean field approximation that seems out of place.  Section 3 clearly states that the posterior on theta is approximated with a posterior on lambda, and this cannot factorize over the dimensions of theta.  Finally, the authors again confuse the posterior on weights with a posterior on distributions of weights in Eq 11.   \tilde{U} is introduced as a function of lambda in Eq 14 and then called with f in line 4 of Alg. 1.  These two types are not interchangeable.

These inconsistencies cast doubt on the subsequent experiments.  Assuming the algorithm is correct, a fundamental experiment is still missing.
To justify this approach, the authors should show how the posterior approximation quality varies as a function of the size of the low dimensional approximation, D.

I reiterate that the idea of approximating the posterior distribution over neural network weights with a posterior distribution over a lower dimensional representation of weights is interesting.  Unfortunately, the abundance of errors in presentation cloud the positive contributions of this paper.

---

### Meta-Review · Area_Chair1 · 2018-12-13
**Interesting idea but has significant technical flaws and lacks clarity**

**Confidence:** 5
**Recommendation:** Reject

**Metareview:**

This paper addresses a promising and challenging idea in Bayesian deep learning, namely thinking about distributions over functions rather than distributions over parameters.  This is formulated by doing MCMC in a functional space rather than directly in the parameter space.  The reviewers were unfortunately not convinced by the approach citing a variety of technical flaws, a lack of clarity of exposition and critical experiments.  In general, it seems that the motivation of the paper is compelling and the idea promising, but perhaps the paper was hastily written before the ideas were fully developed and comprehensive experiments could be run.  Hopefully the reviewer feedback will be helpful to further develop the work and lead to a future submission.

Note: Unfortunately one review was too short to be informative.  However, fortunately the other two reviews were sufficiently thorough to provide enough signal.